# The *Umbelopsis ramanniana* Sensu Lato Consists of Five Cryptic Species

**DOI:** 10.3390/jof8090895

**Published:** 2022-08-23

**Authors:** Ya-Ning Wang, Xiao-Yong Liu, Ru-Yong Zheng

**Affiliations:** 1State Key Laboratory of Mycology, Institute of Microbiology, Chinese Academy of Sciences, Beijing 100101, China; 2College of Life Sciences, Shandong Normal University, Jinan 250014, China

**Keywords:** *Mucoromycota*, *Umbelopsidales*, *Umbelopsidaceae*, five new species, taxonomy, molecular phylogeny, maximum growth temperature test

## Abstract

*Umbelopsis ramanniana* is one of the most commonly reported species within the genus and an important oleaginous fungus. The morphology of the species varies remarkably in sporangiospores, columellae and chlamydospores. However, phylogenetic analyses based on ITS and nLSU rDNA had previously shown insufficiency in achieving species level identification in the genus *Umbelopsis*. In this study, by applying a polyphasic approach involving multi-gene (nSSU, ITS, nLSU, *act1*, MCM7 and *cox1*) phylogeny, morphology and maximum growth temperature, *U. ramanniana* sensu lato was revealed as a polyphyletic group and resolved with five novel taxa, namely *U**. curvata*, *U. dura*, *U. macrospora*, *U. microsporangia* and *U. oblongielliptica*. Additionally, a key for all currently accepted species in *Umbelopsis* was also updated.

## 1. Introduction

*Umbelopsis ramanniana*, a widespread species in the genus *Umbelopsis*, is a promising oleaginous fungus in biochemistry and biotechnology. The species is well-known for accumulating large amounts of lipids, which makes the species useful in studying the mechanism of lipid biosynthesis [1,2,3] and for the biotransformation of oil [4,5]. Ecologically, *U. ramanniana* is a typical inhabitant of forest soils [6] and important in biological rehabilitation [7,8,9,10,11]; it is also frequently isolated from rhizospheres of forest plants [12], or as an endophyte of plants [13].

*Umbelopsis ramanniana* was first described as *Mucor ramannianus* by Möller [14]. Historically, there has been a long debate about the attribution of the species and position of the genus *Umbelopsis* in *Mucorales* [15]. For its similarity with *Mortierella isabellina* (basionym of *U. isabellina*), Linnermann [16] transferred the species to *Mortierella* sect. *Pusilla* (which was revised to sect. *Isabellina* by the author in 1969, nom. inval.). Mil’ko [17], however, retained the species in the genus *Mucor* and introduced a new section *Ramannianus*. Gams [18] maintained it as a species of *Mortierella* and proposed the subgenus *Micromucor*. Von Arx [19] elevated the subgenus to a genus rank as *Micromucor* in *Mucoraceae* and treated the species as *Micromucor ramannianus*. Meyer and Gams [20] combined the genera *Umbelopsis* and *Micromucor* based on the results of restriction fragment length polymorphism data of the whole nuclear ribosomal internal transcribed spacer (ITS) region and phylogenetic reconstruction of ITS1; and, therefore, the species was recombined as its current name *U. ramanniana*.

The species was first reported from pine mycorrhizas collected in Bavaria and Mark Brandenburg of Germany and was proposed with a poorly informative description, i.e., roseate colony, roundish to elongated sporangiospores and two types of chlamydospores [14]. Because of the lack of type information or illustrations, remarkable variations in morphology and biochemistry among strains assigned to this species have been observed in subsequent studies. Consequently, some members with specific features were separated from *U. ramanniana*. For strains with angular sporangiospores other than ellipsoidal ones in the autonym variety *ramanniana*, Linnemann [16] introduced a variety *angulispora*, which was accepted by Chalabuda [21]. Sugiyama et al. [15] found that the ex-neotype strain of the variety *angulispora* should be identified as *U. vinacea* and introduced a new species *U. angularis* for the isolates with angular sporangiospores in the sense of Linnemann. Evans [22] reported that *U. ramanniana* varied significantly across strains under different cultural conditions and described another variety *autotrophica* for the strains showing thiamine independence, pale congo-pink colonies and globose sporangiospores as opposed to the thiamine dependence, darker colonies and ellipsoidal sporangiospores of the variety *ramanniana*. This variety was raised to a species rank as *U. autotrophica* by Meyer and Gams [20]. A third variety *incrustacea* was proposed for strains differing from the original variety in the shape and size of sporangiophores, columellae and sporangiospores [21]. This variety is now a doubtful name due to the lack of related strains all over the world [23].

In addition to the variations exhibited by the varieties of *U. ramanniana* as mentioned above, more morphological, molecular and biochemical variations within the species have been reported. Turner [24] showed that a culture of this species differed from other members in colony color, sporangiophore length, sporangiospore size and chlamydospore shape. Peberdy and Turner [25] pointed out the esterase patterns of different strains varied widely in *U. ramanniana*, which, however, have little or no correlation with the morphological variations. The genetic divergences on ITS and nuclear large subunit (nLSU) rDNA sequences and chromosomal number and size were observed among isolates of *U. ramanniana* [15,20,26]. Based on a more comprehensive investigation on variations of sporangiospores, columellae, chlamydospores and sporangiophores, as well as ITS and nLSU rDNA sequences, Ogawa et al. [27,28] proposed that isolates traditionally identified as *U. ramanniana* were polyphyletic and included at least three morphologically and genetically divergent groups. However, due to little correlation between nuclear rDNA sequences and morphological characteristics, and the lack of robust molecular phylogenies, *U. ramanniana* has not been redefined and the isolates traditionally identified as this species have not been reclassified.

The present work focuses on re-examining the variation within the species complex and determining the taxonomical status of the cultures in *U. ramanniana* sensu lato. We collected many more strains of *U. ramanniana*, especially some well-known ones, such as NRRL 1296, NRRL 5844 and CBS 219.47; performed a comprehensive phylogenetic analysis based on the sequences from six loci; and examined the morphological characters and maximum growth temperatures of the strains. Consequently, a revised circumscription for *U. ramanniana* is presented here and five new species are recognized from the species complex. Furthermore, combined with our previous studies on *Umbelopsis*, a key to known species in this genus is also updated.

## 2. Materials and Methods

### 2.1. Cultures and Isolation

Details of materials studied are listed in Appendix A. Strains from China were isolated by Chen [29] and Wang et al. [23] using the method of Zheng et al. [30] and preserved in the China General Microbial Culture Collection centre, Institute of Microbiology, Chinese Academy of Sciences, Beijing, China (CGMCC) and the State Key Laboratory of Mycology, IM, CAS (Um). Others were obtained from the USDA Agricultural Research Culture Collection, Peoria, Illinois, USA (NRRL) and the Westerdijk Fungal Biodiversity Institute, Utrecht, the Netherlands (CBS). Dried cultures for holotypes were deposited in the Herbarium Mycologicum Academiae Sinicae, IM, CAS, Beijing, China (HMAS).

### 2.2. Media, Cultivation and Morphological Observation

For DNA extraction, strains were cultivated in malt extract (ME: malt extract 2%, peptone 0.1%, and dextrose 2%) for 4–8 days at 20 °C. Isolates were cultivated at 18 °C for 7–14 days on malt extract agar (MEA: malt extract 2% and agar 2%) and cornmeal agar (CMA: cornmeal 2% and agar 2% agar) under natural light for morphophysiological studies [23]. To determine the maximum growth temperature, each strain was tested three times on PDA for 5 days between 25 and 45 °C.

Microscopic observations were conducted with a Zeiss AX10 Imager A2 light microscope using differential interference contrast illumination. Water or Shear’s mounting medium was used for microscopic observation. Description of the sporangial state was based on an integrative observation of all strains within a certain taxon. Capitalized color designations in the descriptions were from Ridgway [31].

### 2.3. DNA Extraction, Amplification and Sequence Analyses

Total genomic DNA extraction, amplification and sequencing of partial nuclear small subunit (nSSU) rDNA, ITS and D1–D3 region of nLSU rDNA, and the partial γ-actin gene (*act1*) were conducted according to the protocols described by Wang et al. [23,32]. The partial regions of DNA replication licensing factor (MCM7) and mitochondrial cytochrome c oxidase subunit 1 (*cox1*) were amplified using the primer pairs Mcm7-8af (or MCM-709f)/MCM7-16r [33,34] and cox1/cox4 [35], respectively. Polymerase chain reaction (PCR) program of the above two loci included an initial denaturation at 94 °C for 5 min, 39 cycles of 94 °C for 1 min, 53 °C for 1 min and 72 °C for 50 s and a final extension of 72 °C for 10 min. DNA sequencing was performed at Majorbio Bio-technology Company Ltd. (Beijing, China) with the PCR primers. Generated sequences were assembled for consensus in Sequencher 4.1.4 (Gene Codes Corp., Ann Arbor, Michigan); then, they were aligned with MAFFT 6.952 [36,37] and optimized manually in BioEdit 7.1.3.0 [38]. Sequence data generated in this study are deposited in GenBank (Appendix A).

Phylogenetic analysis based on the nLSU rDNA sequences was performed by using the neighbor-joining (NJ) method executed in MEGA7 [39] with Kimura 2-parameter model. The topology of the trees was assessed by 1000 bootstrap replications. For multi-gene phylogenetic analyses, optimized sequence alignments of nSSU, ITS, nLSU, *act1*, MCM7 and *cox1* were combined with SequenceMatrix1.7.8 [40]. Phylogenetic analyses using the maximum likelihood (ML), maximum parsimony (MP) and strict clock Bayesian inference (BI) algorithms were performed with RAxML8.0.23 [41], MEGA7 [39] and MrBayes v. 3.0b4 [42,43], respectively. The parameters for ML, MP and BI analyses were set following the methods described by Wang et al. [23,32]. Trees were visualized with Figtree [44] and edited in Adobe Illustrator CS4. The node reliability was assessed by no less than 70% of maximum likelihood bootstrap proportion (MLBP) and maximum parsimony bootstrap support value (MPBS) and no less than 95% of Bayesian posterior probability values (BPP) [45].

## 3. Results

### 3.1. Molecular Phylogenetic Analyses

In order to investigate the phylogenetic relationships of the strains sequenced in this study with the three subclades recognized by Ogawa et al. [28], the nLSU rDNA sequences of the *U. ramanniana* and related species determined by Ogawa et al. [28] were retrieved from GenBank and integrated with nLSU sequences of the strains determined in this study for a phylogenetic analysis. The phylogenetic tree (Appendix A) shows that the strains identified as *U. ramanniana* are polyphyletic and grouped together with *U. angularis*, *U. gibberispora*, *U. heterosporus*, *U. swartii*, and *U. westeae* and *U. wiegerinckiae*, with high bootstrap values. When more strains were added, the three subclades designated by Ogawa et al. [28] were not resolved as three well-supported linages (Appendix A). Six *U. ramanniana* strains sequenced in this study (CGMCC 3.15777–3.15781 and NRRL 1296) were grouped in a clade together with the strains of subclade I. This clade also includes *U. swartii*, and *U. westeae*, but with low bootstrap support (<50%). The strains of the subclade II and subclade III of Ogawa et al. [28] and 13 new isolates employed in this study formed a well-supported (96%) clade together with *U. angularis*, *U. gibberispora*, *U. heterosporus* and *U. wiegerinckiae*. As mentioned by Ogawa et al. [28], the results indicate that strains of the *U. ramanniana* complex represent an assemblage of several genetically distinct species, but nLSU rDNA alone is insufficient in resolving the relationships of the strains in this species complex. Therefore, a phylogenetic construction based on multiple genes was performed in this study to clarify cryptic species in the *U. ramanniana* complex.

Multi-gene phylogenetic analyses were carried out based on the sequence data of six loci generated in this study or retrieved from GenBank (Appendix A). A total of 86 isolates representing all currently accepted taxa of *Umbelopsis* were analyzed with *Mortierella minutissima* and *M. verticillata* as outgroups. Six sequence alignments (including gaps) were obtained with 1667 characters in nSSU, 723 in ITS, 1013 in nLSU, 820 in *act1*, 1032 in MCM7 and 1742 in *cox1*, respectively. The final combined sequence matrix consists of 6996 characters, including 4915 constant, 777 parsimony uninformative and 1304 parsimony informative characters. The most parsimonious tree resulted in a tree length of 4454 steps, consistency index (CI) of 0.655, retention index (RI) of 0.929 and rescaled consistency index (RC) of 0.608. For the Bayesian inference, the GTR + I + G model was selected for nSSU, *act1* and MCM7, GTR + G model for ITS and nLSU and HKY + G model for *cox1,* respectively. The ML, MP and BI analyses produced similar trees with nodes supported by high bootstrap values. The BI phylogeny tree (Figure 1) is presented with BI, ML and MP bootstrap values indicated along branches.

As shown in Figure 1, the strains of the *U. ramanniana* complex (highlighted in green and blue) can be divided into six well-supported clades (C1–6, Figure 1). Clades C1 and C2 were grouped together with *U. angularis*, *U. gibberispora*, *U. heterosporus* and *U. wiegerinckiae*, forming a highly supported lineage (BPP: 1.00, MLBP: 99, MPBS: 100). Clades C3 to C6 were grouped together with *U. swartii* and *U. westeae*, forming another highly supported lineage (BPP: 1.00, MLBP: 97, MPBS: 100). The clade C1 (BPP: 0.98, MLBP: 80, MPBS < 80), including three isolates (CBS 219.47, CGMCC 3.6647 and CGMCC 3.6648), constituted a grade with *U. heterosporus* and *U. wiegerinckiae*. The clade C2 (BPP: 1.00, MLBP: 100, MPBS: 100), including 11 strains, formed a sister clade to *U. angularis*. Four strains (CGMCC 3.15769, CGMCC 3.15770, CGMCC 3.15771 and CGMCC 3.15782) in the clade C3 formed a strongly supported clade (BPP: 1.00, MLBP: 100, MPBS: 100) in both six-genes and nLSU trees. This clade was grouped together with clades C4 to C6 in six-genes tree (Figure 1), but more closely related to the lineage including clades C1 and C2 in nLSU tree (Appendix A). Although five isolates (CGMCC 3.15777–3.15781) in clade C4 (BPP: 1.00, MLBP: 100, MPBS: 100) and strains NRRL 1296 and NRRL 5844 formed a monophyletic group (BPP: 1.00, MLBP: 100, MPBS: 100) in the six-genes tree, they were divided into three clades for their huge variations in morphology (Figure 1, Table 1) and divergencies in some gene makers. All strains of the clade C4 have an insertion of 1160 bp in the mitochondrial *cox1* gene in comparison to NRRL 1296 and NRRL 5844, as well as other *Umbelopsis* species. This indicates that the clade C4 may possess a unique evolutionary pattern in mitochondrial genes. The two clades represented by strains NRRL 1296 and NRRL 5844 contained one strain each. In the nLSU analysis (Appendix A), the strains NRRL 5844 and YODK 004 formed a strongly supported (84%) clade and were diverged distantly from the cluster including NRRL 1296 and clade C4.

### 3.2. Maximum Growth Temperature

A total of 25 strains in the *Umbelopsis ramanniana* species complex were tested three times for their maximum growth temperature. Detailed results for each culture are presented in Appendix A. The maximum growth temperature of those strains varies from 31 °C to 38 °C, which indicates that there may be cryptic species in the species complex. According to above mentioned phylogenetic clades, the maximum growth temperature ranges for the clades C1 to C6 (Figure 1) are 35–36 °C, 33–35 (–36) °C, 31–32 °C, 35 °C, 37 °C and 38 °C, respectively.

### 3.3. Morphology and Taxonomy

Like in other species of the *Umbelopisis*, morphology is an important basis for the taxonomy of the *U. ramanniana* species complex. The following characteristics are of prime importance to the classification of *Umbelopsis*: colony color, the pattern and length of branches, the type and shape of sporangia, the shape and size of columellae and sporangiospores, and the formation of chlamydospores. In this study, more criteria, such as colony diameter, the deliquescence of sporangial walls and the possession of collars, have been adopted to investigate differences in the *U. ramanniana* complex. Combined with the results of phylogenetic analyses, the mainly morphological comparison is summarized in Table 1.

Based on the results of the multi-gene phylogeny (Figure 1), morphological comparison (Table 1) and maximum growth temperature test (Appendix A), five new species are introduced for the isolates formerly identified as *U. ramanniana*. These five novel taxa are described here. Meanwhile, *U. ramannianus* is re-described based on our isolated specimens. Additionally, accompanied by these new members, a diagnostic key for all taxa of *Umbelopsis* is updated herein.

#### 3.3.1. *Umbelopsis curvata* Y.N. Wang, X.Y. Liu and R.Y. Zheng, sp. nov.

Fungal Name: FN570527.

Type: The Netherlands, on *Lactarius deliciosus*, 1947, A. L. van Beverwijk. Holotype HMAS 247509, ex-holotype culture CBS 219.47.

Etymology: *curvata* referring to the shape of sporangiospores, often curving to one side.

Diagnosis: *Umbelopsis curvata* (Figure 2) differs from other species by forming ellipsoidal and often curved sporangiospores with (2.7–) 3.2–4.5 (–5.7) × (1.6–) 2.0–2.4 (–2.8) µm.

Description: Colonies on MEA reaching 50–60 mm in diam. after 10 days at 18 °C, 2–3 mm high, velvety, slightly zonate, not wrinkled, sometimes forming sectors, at first white, then becoming brownish vinaceous to light russet-vinaceous (Ridgway, Pl. XXXIX) because of abundant sporulation on the obverse side, pinkish cinnamon to cinnamon (Ridgway, Pl. XXIX) on the reverse side; on CMA, reaching 43–48 mm in diam. after 10 days at 18 °C, low, much sparser than on MEA, not zonate, slightly russet, substrate mycelia dense. Sporangiophores abundant, hyaline, smooth, compactly sympodial branching from slightly swollen stalks, (100–) 260–670 (–1000) µm long, (3.5–) 4.0–6.0 (–6.5) µm wide near the base and 2.0–3.5 (–4.0) µm wide near the tip, 2–4 (–6) septate, the uppermost septum at (16.5–) 20–25.5 (–28.5) µm below the columella, often one septum near the base. Sporangia globose, (11.0–) 15.5–22.5 (–24.5) µm in diam., reddish-brown, multi-spored, walls smooth and quickly deliquescent leaving small but conspicuous collars. Columellae hyaline, smooth, small but distinct, mostly depressed globose and (4.0–) 6.0–8.5 (–10) × (3.5–) 5.0–7.0 (–8.5) µm, sometimes subglobose and 4.0–6.0 (–7.0) µm diam. Sporangiospores smooth-walled, ellipsoidal and often curving to one side, (2.7–) 3.2–4.5 (–5.7) × (1.6–) 2.0–2.4 (–2.8) µm, reddish in mass, with small oil droplets. Chlamydospores in substrate hyphae, smooth, containing oil droplets, two types of size: macro-chlamydospores abundant on CMA and MEA, globose to subglobose, (20.0–) 23.5–40.0 (–52.0) µm in diam., yellowish brown, usually intercalary and sometimes at the base of sporangiophore branches, solitary or in short chains; micro-chlamydospores less abundant, subglobose, (4.0–) 5.5–8.0 (–10.5) µm in diam., hyaline, terminal, solitary or in mass. Zygospores unknown.

Maximum growth temperature: 35–36 °C.

Additional strains examined: China, Hubei, Shennongjia, Sister Mountain, dung, 2 Aug 1984, 366a (CGMCC 3.6647) and 373a (CGMCC 3.6648).

Notes: The ex-type culture CBS 219.47 of *U. curvata* was originally identified as *U. ramanniana* by Domsch et al. [6] and then illustrated by Meyer and Gams [20]. According to the nLSU phylogeny by Ogawa et al. [28], this culture belonged to subclades III of the *U. ramanniana* complex. In the present study, multi-gene phylogenetic analyses revealed that *U. curvata* was distant from the cluster consisting of *U. angularis* and *U. ramannina* (Figure 1) and formed a sister cluster to a group that contained *U. heterosporus* and *U. wiegerinckiae*. Morphologically, *U. curvata* differs from *U. ramanniana* by forming sporangiospores that are larger (3.2–4.5 × 2.0–2.4 µm) and curved; and can be distinguished from *U. angularis* by not exhibiting angular sporangiospores. Compared to irregular sporangiospores and columellae in *U. heterosporus*, *U. curvata* possesses distinct and depressed globose columella and smooth and ellipsoidal sporangiospores. *Umbelopsis curvata* can be distinguished from *U. wiegerinckiae* by its pinkish colony and sympodial branching. Additionally, *U. curvata* is somewhat similar to *U. macrospora* and *U. oblongielliptica* in the shape of sporangiospores (ellipsoidal and curving to one side; Table 1), but they are distantly related in molecular phylogeny (Figure 1).

#### 3.3.2. *Umbelopsis dura* Y.N. Wang, X.Y. Liu and R.Y. Zheng, sp. nov.

Fungal Name: FN570497.

Type: China, Jilin, Changbai Mountain, forest soil mix, 42°10′548″–44°02′274″ N, 126°35′399″–128°55′815″ E, 360–2654 m alt., Jun 2011, Xiao-yong Liu 12448. Holotype HMAS 247505, ex-holotype culture CGMCC 3.15777.

Etymology: *dura* referring to the permanent nature of sporangial walls, which dissolve slowly and appear stronger than the other species in the *U. ramanniana* complex.

Diagnosis: *Umbelopsis dura* (Figure 3) differs from other species due to slowly deliquescent sporangial walls and Indian red to dark vinaceous-brown colonial color.

Description: Colonies on MEA reaching 50–52 mm in diam. after 10 days at 18 °C, 1 mm high, velvety, zonate, not wrinkled, sometimes forming sectors, at first white, then becoming Indian red (Ridgway, Pl. XXVII) to dark vinaceous-brown (Ridgway, Pl. XXXIX) because of abundant sporulation on the obverse side, pinkish cinnamon to sayal brown (Ridgway, Pl. XXIX) on the reverse side; on CMA, reaching 41–45 mm in diam. after 10 days at 18 °C, low, much sparser than on MEA, zonate, slightly russet, substrate mycelia dense. Sporangiophores abundant, hyaline, smooth, compactly sympodial branching from slightly swollen stalks, (80–) 166–315 (–600) µm long, (2.0–) 4.0–6.0 (–8.0) µm wide near the base and (1.5–) 2.0–4.0 (–5.0) µm wide near the tip, 2–3 (–4) septate, the uppermost septum at (20.0–) 30.5–37.5 (–50.0) µm below the columella, often one septum near the base. Sporangia globose, (10.0–) 15.0–23.5 (–31.5) µm in diam., reddish-brown, multi-spored, walls smooth and slowly deliquescent leaving small or no collars. Columellae hyaline, smooth, small but distinct, depressed globose and (3.5–) 4.0–8.0 (–10.0) × (2.5–) 3.0–7.0 (–9.0) µm, subglobose and (3.0–) 4.0–6.0 (–8.0) µm in diam., or roundish conical and (2.5–) 3.5–6.0 (–7.0) × (3.5–) 4.5–7.0 (–8.0) µm. Sporangiospores smooth-walled, ovoid and sometimes slightly narrowing on one end, (2.4–) 2.8–3.3 (–4.6) × (1.6–) 1.9–2.4 (–2.8) µm, reddish in mass, with small oil droplets. Chlamydospores in substrate hyphae, smooth, containing oil droplets, two types of size: macro-chlamydospores undiscovered on CMA and less abundant on MEA, globose to subglobose, (14.0–) 16.0–23.5 (–27.5) µm in diam., yellowish brown, intercalary, solitary; micro-chlamydospores abundant, subglobose, (3.0–) 5.0–7.5 µm in diam., hyaline, terminal, single or in mass. Zygospores unknown.

Maximum growth temperature: 35 °C.

Additional strains examined: China, Jilin, Changbai Mountain, forest soil mix, 42°10′548″–44°02′274″ N, 126°35′399″–128°55′815″ E, 360–2654 m alt. Jun 2011, Xiao-yong Liu 12,448 CGMCC 3.15778, CGMCC 3.15779, CGMCC 3.15780 and CGMCC 3.15781.

Notes: *Umbelopsis dura* can be separated from all other *Umbelopsis* species in the *cox1* gene by a significant insertion. Phylogenetic inferences show that this species is closely related to *U. oblongielliptica*. However, they can be easily distinguished by the characteristics of the growth rate (41–45 mm vs. 60–62 mm in diam. on CMA after 10 days at 18 °C; Table 1), the deliquescence of sporangial walls (slowly vs. quickly) and the shape and size of sporangiospores (ovoid and 2.8–3.3 × 1.9–2.4 µm vs. oblong-ellipsoidal and 4.0–5.0 × 1.6–2.0 µm; Table 1). Although the colony colors of *U. dura* and *U. angularis* are somewhat similar, the two species differ in their sporangiospores shape and are distantly related in molecular phylogeny (Figure 1).

#### 3.3.3. *Umbelopsis macrospora* Y.N. Wang, X.Y. Liu and R.Y. Zheng, sp. nov.

Fungal Name: FN570526.

Type: United Kingdom, associated with *Pinus sylvestris*. Holotype HMAS 247507, ex-holotype culture NRRL 5844.

Etymology: *macrospora* referring to the size of its sporangiospores, bigger than other species in the *U. ramanniana* complex.

Diagnosis: *Umbelopsis macrospora* (Figure 4) is recognized by forming oblong-ellipsoidal to ellipsoidal sporangiospores with (3.6–) 4.0–4.9 (–5.5) × (1.7–) 2.4–2.8 (–3.2) µm. In addition, it is similar to *U. oblongielliptica*, but differs in the absence of macro-chlamydospore and the growth rate for colonies.

Description: Colonies on MEA reaching 62 mm in diam. after 10 days at 18 °C, 1–2 mm high, velvety, zonate, not wrinkled, without sectors, at first white, then becoming russet-vinaceous to vinaceous-brown (Ridgway, Pl. XXXIX) because of abundant sporulation on the obverse side, pinkish cinnamon to cinnamon (Ridgway, Pl. XXIX) on the reverse side; on CMA, reaching 46 mm in diam. after 10 days at 18 °C, low, much sparser than on MEA, not zonate, slightly russet, substrate mycelia dense. Sporangiophores abundant, hyaline, smooth, compactly sympodial branching from slightly swollen stalks, (100–) 215–685 (–900) µm long, (3.5–) 4.3–6.3 (–7.5) µm wide near the base and 2.0–4.0 µm wide near the tip, 3–4 (–6) septate, the uppermost septum at 24.5–39.5 µm below the columella, one septum near the base. Sporangia globose, (13.5–) 16.0–23.0 (–25.5) µm in diam., reddish-brown, multi-spored, walls smooth and quickly deliquescent leaving small but conspicuous collars. Columellae hyaline, smooth, small but distinct, mostly depressed globose and (4.0–) 6.0–8.0 (–10) × (3.5–) 4.0–6.0 (–7.0) µm, sometimes subglobose and (3.0–) 5.0–7.0 (–8.0) µm in diam. Sporangiospores smooth-walled, oblong-ellipsoidal to ellipsoidal and sometimes slightly curving to one side, (3.6–) 4.0–4.9 (–5.5) × (1.7–) 2.4–2.8 (–3.2) µm, reddish in mass, with or without small oil droplets. Chlamydospores in substrate hyphae, smooth, containing oil droplets, one type of size: macro-chlamydospore undiscovered on CMA or MEA; micro-chlamydospores abundant, subglobose, (3.0–) 6.0–8.0 (–10.0) µm in diam., hyaline, terminal, single or in mass. Zygospores unknown.

Maximum growth temperature: 38 °C.

Notes: The only strain NRRL 5844 of *U. macrospora* was received as *U. ramanniana*. Previously, the strains NRRL 5884 and NRRL 1296 (the ex-hotype of *U. oblongielliptica*) were widely used as *U. ramanniana* in the phylogenetic analyses. However, Nagy et al. [26] reported that the chromosomal banding patterns in those two cultures are significantly diverse in orthogonal field alternation gel electrophoreses (OFAGE) and contour clamped homogeneous electric field gel electrophoreses (CHEF). In the present study, multi-gene phylogenetic inferences show that *U. macrospora* is basal to the *U. oblongielliptica*/*U. dura* cluster (BPP: 1.00; MLBP: 94; MPBS: 100). However, it was separated from this cluster in the nLSU tree (Appendix A). This species is morphologically similar to *U. oblongielliptica* but differs in the growth rate (46 mm vs. 60–62 mm in diam. after 10 days at 18 °C), sporangiospore size (4.0–4.9 × 2.4–2.8 µm vs. 4.0–5.0 × 1.6–2.0 µm) and macro-chlamydospores (undiscovered vs. abundant). Considering the significant divergences in the nLSU tree (Appendix A), morphological characteristics and chromosomal banding patterns demonstrated by Nagy et al. [26], those two isolates were treated as different species.

According to the results of nLSU phylogeny (Appendix A), strain YODK 004 seems like a second specimen of *U. macrospora*. However, the sporangiospores size of this culture is 2.2 ± 0.05 × 1.6 ± 0.05 µm, which is smaller than NRRL 5844 (Ogawa et al. 2005). To determine whether this strain belongs to the species, it is necessary to further confirm its morphological characteristics and sequences of other gene loci.

#### 3.3.4. *Umbelopsis microsporangia* Y.N. Wang, X.Y. Liu and R.Y. Zheng, sp. nov.

Fungal Name: FN570525.

Type: China, Jilin, Changbai Mountain, forest soil mix, 42°10′548″–44°02′274″ N, 126°35′399″–128°55′815″ E, 360–2654 m alt., Jun 2011, Xiao-yong 12448. Holotype HMAS 247506, ex-holotype culture CGMCC 3.15769.

Etymology: *microsporangia* referring to its smaller sporangia than other species in the *U. ramanniana* complex.

Diagnosis: *Umbelopsis microsporangia* (Figure 5) differs from other species due to a relatively low maximum growth temperature of 31–32 °C. It is also characterized by smaller sporangia (10.0–) 14.0–18.5 (–22.5) µm.

Description: Colonies on MEA reaching 50–55 mm in diam. after 10 days at 18 °C, 1 mm high, velvety, slightly zonate, not wrinkled, often forming sectors, at first white, then becoming brownish vinaceous (Ridgway, Pl. XXXIX) because of abundant sporulation on the obverse side, light pinkish cinnamon to pinkish cinnamon (Ridgway, Pl. XXIX) on the reverse side; on CMA, reaching 35–40 mm in diam. after 10 days at 18 °C, low, much sparser than on MEA, not zonate, slightly russet, substrate mycelia dense. Sporangiophores abundant, hyaline, smooth, compactly sympodial branching from slightly swollen stalks, (80–) 135–510 (–745) µm long, (2.0–) 3.5–5.0 (–7.0) µm wide near the base and (1.5–) 2.0–3.5 µm wide near the tip, 2–3 (–4) septate, the uppermost septum at (17.5–) 21.0–32.0 (–39.0) µm below the columella, one septum near the base. Sporangia globose, (10.0–) 14.0–18.5 (–22.5) µm in diam., reddish-brown, multi-spored, walls smooth and quickly deliquescent leaving small or no collars. Columellae hyaline, smooth, small but distinct, mostly depressed globose and (3.0–) 4.0–6.5 (–10.0) × (2.5–) 3.5–5.5 (–9.0) µm, sometimes subglobose and (2.5–) 4.0–6.0 µm in diam. Sporangiospores smooth-walled, ovoid to subglobose, uniform in size, (2.4–) 2.6–3.2 (–4.0) × 1.8–2.5 (–2.8) µm, reddish in mass, smooth-walled, without small oil droplets. Chlamydospores in substrate hyphae, smooth, containing oil droplets, two types of size: macro-chlamydospores rare on CMA and less abundant on MEA, globose to subglobose, (9.0–) 16.0–26.0 (–39.5) µm in diam., yellowish brown, usually intercalary and sometimes at the base of sporangiophore branches, solitary; micro-chlamydospores abundant, subglobose, (3.0–) 6.0–8.5 (–12.5) µm in diam., hyaline, terminal, single or in mass. Zygospores unknown.

Maximum growth temperature: 31–32 °C.

Additional strains examined: China, Jilin, Changbai Mountain, forest soil mix, 42°10′548″–44°02′274″ N, 126°35′399″–128°55′815″ E, 360–2654 m alt. Jun 2011, Xiao-yong Liu 12448. CGMCC 3.15770, CGMCC 3.15771 and CGMCC 3.15782.

Notes: *Umbelopsis microsporangia* formed a strongly supported clade (BPP: 1.00; MLBP: 100%; MPBS: 100%) and occurred as a sister clade of a polytomy consisting of *U. dura*, *U. oblongielliptica* and *U. macrospora* (Figure 1). Morphologically, *U. microsporangia* produces sporangiospores that are smaller than *U. oblongielliptica* and *U. macrospora* (2.6–3.2 × 1.8–2.5 µm vs. 4.0–5.0 × 1.6–2.0 µm and 4.0–4.9 × 2.4–2.8 µm, respectively; Figure 1 and Table 1) and can be distinguished from *U. dura* by its brownish vinaceous colony color and rapidly deliquescent sporangia. Furthermore, the maximum growth temperature range of the *U. microsporangia* is 31–32 °C, which is noticeably lower than those of the above mentioned three species (35–38 °C; Appendix A).

#### 3.3.5. *Umbelopsis oblongielliptica* Y.N. Wang, X.Y. Liu and R.Y. Zheng, sp. nov.

Fungal Name: FN570524.

Type: USA, Wisconsin, substrate unknown. Holotype HMAS 247508, ex-holotype culture NRRL 1296.

Etymology: *oblongielliptica* referring to the shape of sporangiospores.

Diagnosis: *Umbelopsis oblongielliptica* (Figure 6) differs from other species by forming oblong-ellipsoidal sporangiospores (3.2–) 4.0–5.0 (–5.5) × 1.6–2.0 (–2.5) µm. This species is also characterized by the rapid colony extension (up to 64–70 mm in diameter after 10 d at 18 °C on MEA) and the production of abundant macro-chlamydospores on the substrate of both CMA and MEA.

Description: Colonies on MEA reaching 64–70 mm in diam. after 10 days at 18 °C, 1 mm high, velvety, zonate, wrinkled or not, sometimes forming sectors, at first white, then becoming russet-vinaceous (Ridgway, Pl. XXXIX) because of abundant sporulation on the obverse side, pinkish cinnamon to brown (Ridgway, Pl. XXIX) on the reverse side; on CMA, reaching 60–62 mm in diam. after 10 days at 18 °C, low, much sparser than on MEA, not zonate, slightly russet, substrate mycelia dense. Sporangiophores abundant, hyaline, smooth, compactly sympodial branching from slightly swollen stalks, (120–) 270–475 (–980) µm long, 3.5–7.0 µm wide near the base and (1.8–) 2.0–4.0 µm wide near the tip, 3–4 (–6) septate, the uppermost septum at 17.5–27.5 µm below the columella, often one septum near the base. Sporangia globose, (11.5–) 15.5–21.5 (–26.0) µm in diam., reddish-brown, multi-spored, walls smooth and quickly deliquescent leaving small but conspicuous collars. Columellae hyaline, smooth, small but distinct, mostly depressed globose and (4.5–) 6.0–8.5 (–10) × (3.0–) 5.0–7.0 (–8.5) µm, sometimes subglobose and 5.0–7.0 µm in diam., occasionally roundish conical and 3.5–6.0 × 4.0–6.5 µm. Sporangiospores smooth-walled, oblong-ellipsoidal and sometimes slightly curved to one side, (3.2–) 4.0–5.0 (–5.5) × 1.6–2.0 (–2.5) µm, reddish in mass, with or without small oil droplets. Chlamydospores in substrate hyphae, smooth, containing oil droplets, two types of size: macro-chlamydospores abundant on CMA and MEA, globose to subglobose, (15.5–) 28.0–39.5 (–47.5) µm in diam., yellowish brown, intercalary, solitary or in chains; micro-chlamydospores abundant, subglobose, (3.0–) 4.0–8.0 (–13.0) µm in diam., hyaline, terminal, single or in mass. Zygospores unknown.

Maximum growth temperature: 37 °C.

Notes: The only culture NRRL 1296 of *U. oblongielliptica* was received as *U. ramanniana*. It is phylogenetically closely related to *U. dura* (BPP: 1.00; MLBP: 100%; MPBS: 100%) and *U. macrospora* (Figure 1), but morphologically differs from the former species by forming oblong-ellipsoidal sporangiospores (4.0–5.0 × 1.6–2.0 µm; Table 1), and the later one by producing abundant chlamydospores on MEA and narrower sporangiospores (1.6–2.0 µm vs. (1.7–) 2.4–2.8 (–3.2) µm in width; Table 1).

#### 3.3.6. *Umbelopsis ramanniana* (Möller) W. Gams, Mycol. Res. 107(3): 349 (2003)

≡*Mucor ramannianus* Möller, Z. Forst- u. Jagdw. 35: 330 (1903)≡*Mortierella ramanniana* (Möller) Linnem., Mucor.-Gatt. Mortierella Coem.: 19 (1941)≡*Micromucor ramannianus* (Möller) Arx, Sydowia 35: 19 (1984)

Diagnosis: *Umbelopsis ramanniana* (Figure 7) is recognized by brownish vinaceous to light russet-vinaceous colonies, small but distinct columellae and ellipsoidal sporangiospores with 2.4–3.6 (–4.6) × 1.6–2.4 (–2.6) µm.

Description: Colonies on MEA reaching 48–56 mm in diam. after 10 days at 18 °C, 1–2 (–3) mm high, velvety, zonate, not wrinkled, without sectors, at first white, then becoming brownish vinaceous to light russet-vinaceous (Ridgway, Pl. XXXIX) because of abundant sporulation on the obverse side, pinkish cinnamon to cinnamon (Ridgway, Pl. XXIX) on the reverse side; on CMA, reaching 40–44 mm in diam. after 10 days at 18 °C, low, much sparser than on MEA, not zonate, slightly russet, substrate mycelia dense. Sporangiophores abundant, hyaline, smooth, compactly sympodial branching from slightly swollen stalks, (100–) 275–745 (–1275) µm long, (3.0–) 4.0–6.0 (–8.0) µm wide near the base and (1.5–) 2.0–4.0 µm wide near the tip, 2–4 (–7) septate, the uppermost septum at (9.0–) 16.0–26.0 (–37.0) µm below the columella, one septum near the base. Sporangia globose, (11.0–) 15.0–23.0 (–29.5) µm in diam., reddish-brown, multi-spored, walls smooth and quickly deliquescent leaving small but conspicuous collars. Columellae hyaline, smooth, small but distinct, mostly subglobose and (4.0–) 6.0–8.0 (–10.0) µm in diam., sometimes depressed globose and (4.0–) 6.0–10.0 (–13.0) × (3.5) 5.0–8.5 (–12.0) µm, or ovoid to oblong-ovoid and 4.0–6.0 (–8.0) × (4.5–) 5.0–7.0 (–9.5) µm. Sporangiospores smooth-walled, ellipsoidal, 2.4–3.6 (–4.6) × 1.6–2.4 (–2.6) µm, reddish in mass, without small oil droplets. Chlamydospores in substrate hyphae, smooth, containing oil droplets, two types of size: macro-chlamydospores less abundant on CMA or MEA, globose to subglobose, (12.0–) 16.0–30.0 (–44.0) µm in diam., yellowish brown, usually intercalary and seldom at the base of sporangiophore branches, solitary; micro-chlamydospores sparse, subglobose, (3.5–) 6.0–8.0 (–10.0) µm in diam., hyaline, terminal, single. Zygospores unknown.

Maximum growth temperature: 33–35 (–36) °C.

Strains examined: China, Jilin, Changbai Mountain, forest soil mix, 42°10′548″–44°02′274″ N, 126°35′399″–128°55′815″ E, 360–2654 m alt., Jun 2011, Xiao-yong Liu (CGMCC 3.15772, CGMCC 3.15773, CGMCC 3.15774, CGMCC 3.15775 and CGMCC 3.15776); China, Hubei, Shennongjia, Jiuhuping, from Gastrodia elata, 1984, 377a (CGMCC 3.6646); China, Tibet, Lulang, rotten cloth, 5 Aug 2009, Xue-wei Wang (CGMCC 3.15783); China, Hubei, Shennongjia, Nanyinzhai, soil under *Fargesia spathacea*, Jul 1984, 402b (CGMCC 3.16356); China, Fujian, Wuyishan National Nature Reserve, wild fruits fell on the ground, 27°44′414″ N, 117°41′113″ E, 1232 m alt., 20 Jun 2012, Ya-wing Wang 12,954 (CGMCC 3.15784 and CGMCC 3.15785); China, Fujian, Wuyishan City, soil in grove of bamboo, 20 Jun 2012, Ya-ning Wang 12,983 (CGMCC 3.15786).

Notes: The strains of clade C2 in the multi-gene phylogenetic tree (Figure 1) were characterized by producing ellipsoidal spores, small but distinct columellae and two types of chlamydospores, which is compatible with Möller’s description for *U. ramanniana* [14]. Moreover, they formed a sister lineage to *U. angularis*, which was a variant of *U. ramanniana* in the sense of Linnermann [15,16]. As a result, the strains in clade C2 were regarded as representative of the *U. ramanniana* in this study.

*Umbelopisis ramanniana* can be distinguished from *U. angularis* by having ellipsoidal sporangiospores and a lighter colony color. *Umbelopisis curvata* (C1, Figure 1) can be distinguished from *U. ramanniana* (C2, Figure 1) by its columellae shape and the shape and size of sporangiospores (Table 1). In addition, slight morphological variations were observed in several isolates of this clade. The sporangiophores of isolates CGMCC 3.6646 and CGMCC 3.15774 are shorter than other isolates, which may be intraspecific variations caused by differences in geographical environment.

In this species complex, remarkable variations were found in the monophyletic cluster consisting of *U. microsporangia*, *U. dura*, *U. oblongielliptica* and *U. macrospora* (C3, C4, C5 and C6, respectively; Figure 1). *Umbelopisis microsporangia* and *U. dura* produce smaller ovoid sporangiospores; however, some of them are oblong-ellipsoidal to ellipsoidal in *U. oblongielliptica* and *U. macrospora* (Figure 1 and Table 1). Compared to *U. microsporangia* and other taxa in the *U. ramanniana* complex, the deliquescence of sporangia walls in *U. dura* is slower. Moreover, the colony diam. after 10 days at 18 °C in the clade *U. microsporangia* is 35–40 mm, which is 41–45 mm in *U. dura*. Regarding their morphology, two single strain clades C5 and C6 can be easily distinguished from each other by their colony diam., the shape and size of columellae, sporangiospores size and macro-chlamydospores (abundant vs. undiscovered). Therefore, we treated them as different taxa *U. oblongielliptica* and *U. macrospora.*

#### 3.3.7. Key to Species of *Umbelopsis*

1.Sporangia sometimes with neck-like base
*Umbelopsis longicollis*
–Sporangia without neck-like base22.Colonies drab grey when the sporangia are produced heavily3–Colonies roseate to almost white when the sporangia are produced heavily43.Sporangia obovate to ovate
*U. ovata*
–Sporangia globose to subglobose
*U. isabellina*
4.With two kinds of sporangia, single- and multi-spored
*U. dimorpha*
–With one kind of sporangia, single- or multi-spored55.Colony color white to pale pinkish; sporangia single-spored with 4.5–8.0 µm in diam.; sporangiospores globose with the same size of sporangia
*U. nana*
–Colony color roseate to pale pinkish; sporangia multi-spored larger than 10 µm; sporangiospore globose to oblong-ellipsoidal far smaller than sporangia66.Sporangiospores globose, maximum growth temperature 40 °C
*U. autotrophica*
–Sporangiospores subglobose, ovoid to oblong-ellipsoidal, angular or irregular; maximum growth temperature no more than 39 °C77.Sporangiophores no more than 100 µm in length8–Sporangiophores mainly more than 100 µm in length108.Colonies reaching 20–21 mm in diam. on MEA after 7 days at 20 °C; sporangiospores globose, ellipsoidal or angular
*U. sinsidoensis*
–Colonies reaching more than 30 mm in diam. on MEA after 7 days at 20 °C; sporangiospores angular99.Colonies russet; sporangiophores umbellately branched from distinct vesicles on agar surface, (16–) 23–63 (−80) µm in length; columellae small but distinct 3.2–5.5 × 1.6–3.8; without micro-chlamydospores
*U. changbaiensis*
–Colonies light russet-vinaceous pinkish; sporangiophores simply branched from slightly swollen stalks, (27.7–) 47.4–94.8 µm in length; columellae absent or slightly convex up to 1 µm diam.; with micro-chlamydospores
*U. vinacea*
10.Sporangia fusiform; without columella
*U. fusiformis*
–Sporangia globose to subglobose; with distinct columellae1111.Columellae distinct to inconspicuous and variable in shape and size; sporangiospores variable in shape and size up to 3.5–5 × 5–11 µm
*U. heterosporus*
–Columellae distinct and uniform; sporangiospores mostly uniform and no more than 6 µm in diam1212.Sporangiophores mainly umbellately branched from swollen portion of the subtended stalk
*U. wiegerinckiae*
–Sporangiophores sympodial branched from slightly swollen stalks1313.Sporangiospores angular
*U. angularis*
–Sporangiospores not angular1414.Sporangiospores with appendage15–Sporangiospores smooth1715.Sporangiophores straight, sporangiospores without a narrowed end, walls thickened unilaterally
*U. gibberispora*
–Sporangiophores sometimes recurved; sporangiospores with a narrowed end, walls thickened or not1916.Sporangiospores tear-shaped with 4.0–6.0 ×2.0–2.5 µm
*U. swartii*
–Sporangiospores oval to clavate with distinctively thickened wall at the narrower end with 4.0–6.0–8.0 × 2.5–3.5 µm
*U. westeae*
17.Colony color deep roseate, Indian red to dark vinaceous-brown; sporangial walls slowly deliquescent
*U. dura*
–Colony color roseate to pinkish, brownish vinaceous to light russet-vinaceous; sporangial walls quickly deliquescent1818.Sporangiospores ovoid to ellipsoidal no curved, mostly 2.4–3.5 µm in length19–Sporangiospores ellipsoidal to oblong-ellipsoidal and sometimes slightly curved to one side, mainly 3.5–4.9 µm in length2019.Colony diam. on CMA reaching 35–40 mm in diam. after 10 days at 18 °C; maximum growth temperature 31–32 °C; columellae mostly depressed globose (3.0–) 4.0–6.5 (–10.0) × (2.5–) 3.5–5.5 (–9.0) µm
*U. microsporangia*
–Colony diam. on CMA reaching 40–44 mm in diam. after 10 days at 18 °C, maximum growth temperature higher than 32 °C; columellae mostly subglobose (4.0–) 6.0–8.0 (–10.0) µm
*U. ramanniana*
20.Colony diam. on CMA reaching 60–62 mm in diam. after 10 days at 18 °C; colonies 1 mm in high; sporangiophores(120–) 270–475 (–980) µm in length
*U. oblongielliptica*
–Colony diam. on CMA reaching 43–48 mm in diam. after 10 days at 18 °C; colonies up to 2 or 3 mm; sporangiophores commonly more than 500 µm in length2121.Sporangiospores oblong-ellipsoidal to ellipsoidal, (3.6–) 4.0–4.9 (–5.5) × (1.7–) 2.4–2.8 (–3.2) µm; macro-chlamydospore not formed on CMA and MEA
*U. macrospora*
–Sporangiospores ellipsoidal (2.7–) 3.2–4.5 (–5.7) × (1.6–) 2.0–2.4 (–2.8) µm in size; macro-chlamydospore abundant on CMA and MEA
*U. curvata*


## 4. Discussion

The noticeable variations of *Umbelopsis ramanniana* have been described by the morphological characteristics of their sporangiospores shape, columella size and chlamydospore production, and by their ubiquitous ecological distribution by many studies [22,24,27,28]. However, it was difficult to make a clear delimitation as those morphological characteristics vary continuously among the isolates. Previous research indicated that ITS and nLSU rDNA were not sufficient in achieving species-level identification for the genus *Umbelopsis* [15,20,27,28,46]. In the present study, based on a combined analysis of multi-loci phylogeny (nSSU, ITS and nLSU rDNA, *act1*, MCM7 and *cox1*), morphology and maximum growth temperature, isolates previously identified as *U. ramanniana* were re-examined and divided into six clades, including five morphologically cryptic lineages. They are described herein as novel species, namely *U. curvata*, *U. dura*, *U. macrospora, U. microsporangia* and *U. oblongielliptica*.

*Umbelopsis ramanniana* was meagerly described by Möller [14] without illustration or typification. Based on the culture isolated from mycorrhiza of trees in Eberswalde (Brandenburg, Germany; habitation in protolog) by Möller, Lendner [47] provided a more detailed description and illustration of sporangia, columella and chlamydospores for the species. According to the two studies mentioned above, this species produces roseate colonies, ellipsoidal sporangiospores (2.5 × 1.7 µm), small but distinct columellae and two types of chlamydospores. In this study, the isolates in the clade C2 (Figure 1) tend to fit the descriptions of Möller [14] and Lendner [47] best. Moreover, the clade C2 formed a sister linage to *U. angularis*, which is a closely related species to *U. ramanniana* in the sense of Linnemann [16]. Therefore, strains in the clade C2 were chosen as the deputies of *U. ramanniana*, although there may have been some arbitrariness.

Based on the sequences of the nLSU rDNA D1/D2 region, Ogawa et al. [28] examined intraspecific variations of *U. ramanniana* and pointed out that this species is an assemblage of several genetically distinct species. In the present study, *U. ramanniana* was polyphyletic in the nLSU tree (Appendix A), which supports the conclusion of Ogawa et al. [28]. When 23 strains of *U. ramanniana* were added to the subclades identified by Ogawa et al. [28], more divergent clades were recognized in this species complex. However, due to the low bootstrap values in the nLSU tree, the phylogenetic relationships between subclades of *U. ramanniana* and closely related species were undetermined. As mentioned by Ogawa et al. [27,28], it is more reasonable to make the taxonomic treatment when species relationships are clarified.

In the present study, the six-loci phylogeny of *Umbelopsis* was reconstructed and obtained strong support values. The *U. ramanniana* complex is polyphyletic and can be divided into six clades. The first two clades *U. curvata* (C1) and *U. ramanniana* (C2) were located in a well-supported lineage that included *U. angularis*, *U. gibberispora*, *U. heteropsproa* and *U. wiegerinckiae*. The last two species in this lineage are recently reported, of which, *U. wiegerinckiae* is characterized by forming umbellate sporangiophores, and *U. heterosporus* produces varied sporangiophores and columellae [48,49]. The other four species can be distinguished from each other by their unique sporangiospores. In detail, they are the biggest and unilaterally thickened oblong-ellipsoidal sporangiospores in *U. gibberispora*, bigger and oblong-ellipsoidal with slight curved to one side in *U. curvata*, smaller and ellipsoidal without any appendage in *U. ramanniana* and smallest and polygonal in *U. angularis*. Variations of sporangiospores seem to confirm the hypothesis that tight sporangial walls physically limit the free expansion of sporangiospores and consequently several kinds of shape of sporangiospores could evolve [15]. Moreover, it appears that the tighter sporangial walls seem to result in the deeper sporangial and colonial color. The color of colonies is pale vinaceous in *U. gibberispora*, brownish vinaceous to light russet-vinaceous in *U. curvata* and *U. ramanniana,* and etruscan red to prussian red in *U. angularis*. The acquisition of a tight sporangial wall should be a main evolution strategy for taxa in this lineage, which supports the hypothesis that relatively few mutations are required to determine a tight sporangial wall [15]. Hence, the differentiation of the lineage is probably a recent event in the evolution of the *Umbelopsis*. The other four clades *U. dura* (C4), *U. macrospora* (C6), *U. microsporangia* (C3) and *U. oblongiellptica* (C5) were clustered together with *U. swartii* and *U. westeae*. These species formed another well-supported lineage. In this lineage, the sporangiospore shape of the above mentioned six species also varies from appendage to smooth and subglobose to oblong-ellipsoidal. In detail, the sporangiospores are subglobose to ovoid in *U. microsporangia*, ovoid in *U. dura*, ellipsoidal to oblong-ellipsoidal with irregularly slight curved to one side in *U. macrospora* and *U. oblongielliptica,* and appendaged in *U. swartii* and *U. westeae*. It is suggested that variations in sporangiospores shape and size are acquired independently in those two lineages. Moreover, *U. dura* and *U. microsporangia* possess a similar sporangiospore shape and size but differ in their maximum growth temperature. Additionally, the maximum growth temperature increased gradually from 31–32 °C to 38 °C in the clades *U. microsporangia* (C3) to *U. macrospora* (C6) (Appendix A). It is indicated that at least two strategies occurred in the evolution of the lineage of *U. swartii* through to *U. microsporangia.*

As shown in previous studies, the ITS barcode was not sufficient in achieving species level identification in *Umbelopsis* [27,32]. Secondary barcodes, usually protein-coding genes, have been introduced for species discrimination [50,51]. The extensively used loci, such as beta-tubulin (ßtub), translation elongation factor 1-alpha (TEF1), the largest subunit of RNA polymerase II (RPB1) and the second largest subunit of RNA polymerase II (RPB2), were tested, but resulted in multiple copies [32]. The present study preliminarily suggests that the MCM7 should be a suitable secondary barcode, but this needs to be further studied.

## 5. Conclusions

The multi-gene (nSSU, ITS, nLSU, *act1*, MCM7 and *cox1*) phylogeny were proved to be reliable indications of taxon differentiation for the *Umbelopsis ramanniana* complex. Five species are newly described from the species complex: *U. curvata*, *U. dura*, *U. macrospora*, *U. microsporangia* and *U. oblongielliptica*. Those species can be distinguished by morphological traits in combination with the speed of growth and their maximum growth temperature.

## Figures and Tables

**Figure 1 jof-08-00895-f001:**
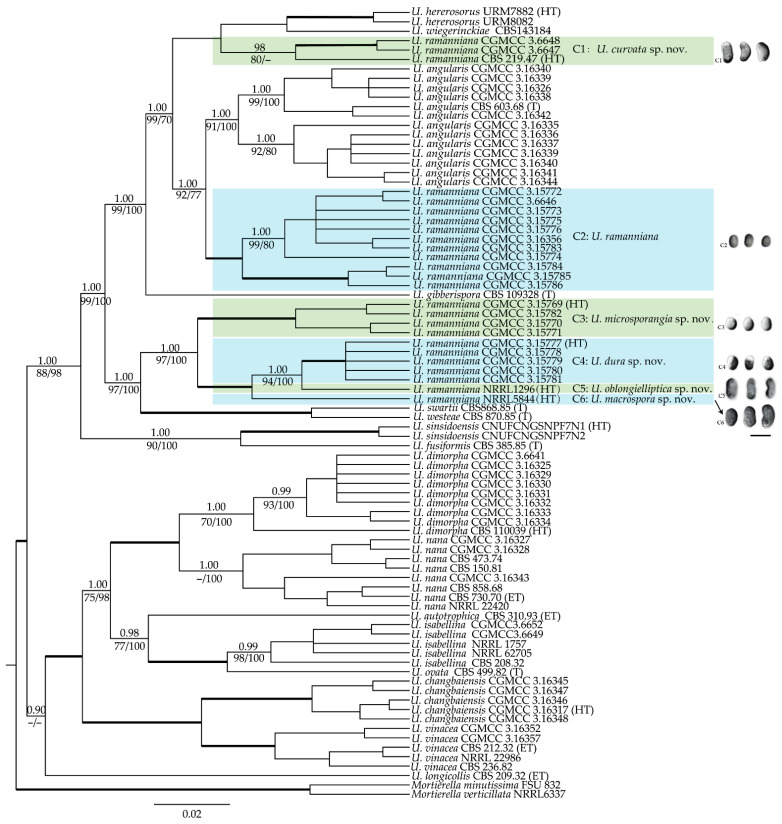
Phylogenetic tree for *Umbelopsis* based on a combined data matrix comprised alignments of nSSU, ITS, nLSU, *act1*, MCM7 and *cox1* generated from Bayesian analyses with *Mortierella* as outgroups. Values above the branches represent significant Bayesian posterior probability values (BPP ≥ 0.95), and values below the branches are maximum likelihood bootstrap proportion (MLBP ≥ 70%) and maximum parsimony bootstrap support values (MPBS ≥ 70%). Branches in bold indicate strong support (MLBP: 100%, MPBS: 100%, BPP: 1.00). Missing or weakly supported nodes (MLBP < 70%, MPBS < 70% or BPP < 0.95) are denoted by a minus sign “−”. The bar at the lower left indicates 0.02 expected changes per site. The new species are highlighted in green and blue. The sporangiospores of novel species established in this study are illustrated on the right side of the tree (scale bar = 5 µm) and correlated with each clade of the *U. ramanniana* complex using the same clade numbers. *U.* = *Umbelopsis*. T = ex-type strain, ET = ex-epitype strain and HT = ex-holotype strain.

**Figure 2 jof-08-00895-f002:**
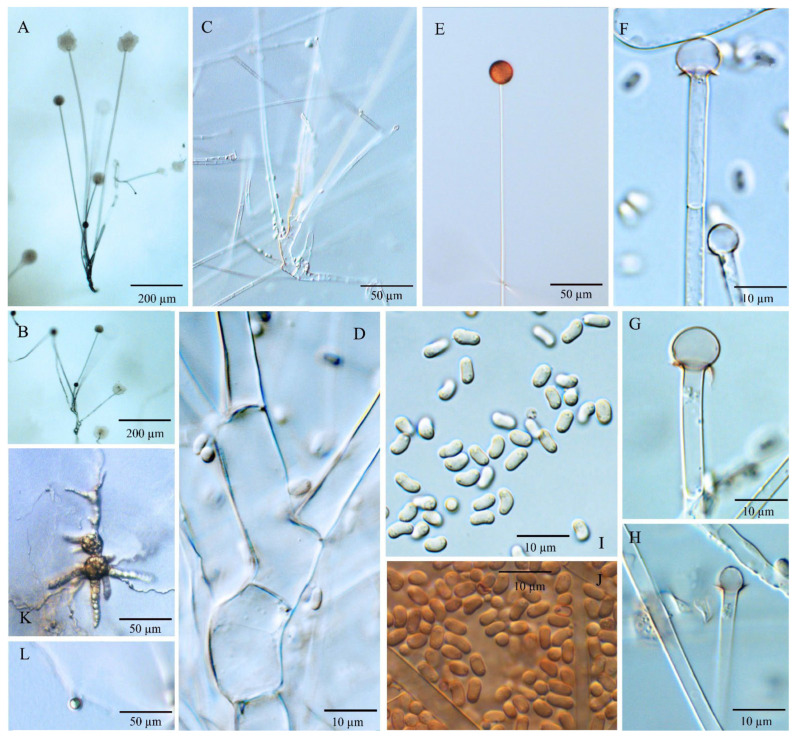
*Umbelopsis curvata*. (**A**–**C**) Branched sporangiophores. (**D**) Branch point of sporangiophore. (**E**) Sporangium at tip of sporangiophore. (**F**–**H**) Various shapes of collars and columellae at sporangiophore tips after the sporangia have been dissolved. (**I**, **J**) Sporangiospores. (**K**) Macro-chlamydospores. (**L**) Micro-chlamydospore.

**Figure 3 jof-08-00895-f003:**
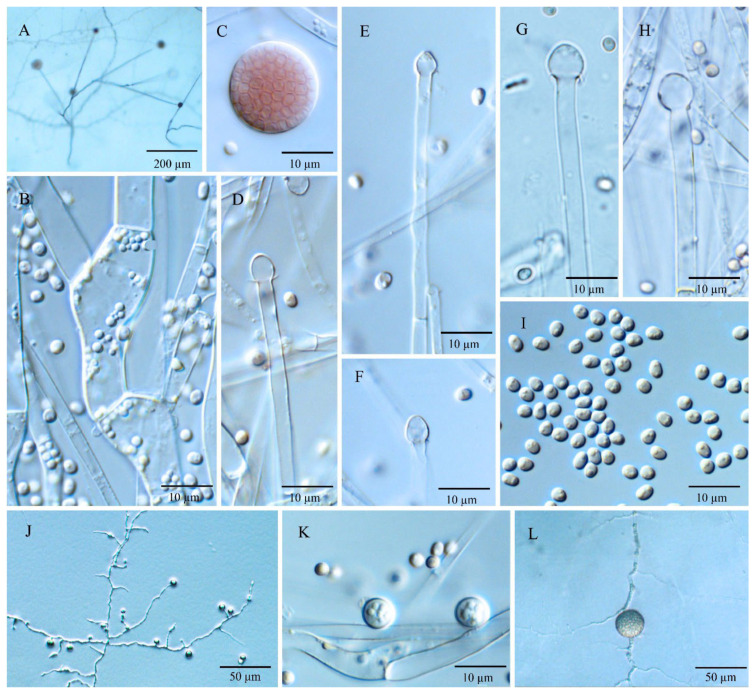
*Umbelopsis dura*. (**A**) The main branching pattern of sporangiophore. (**B**) Branch point of sporangiophore. (**C**) Sporangium at tip of sporangiophore. (**D**–**H**) Various shapes of collars and columellae at sporangiophore tips after the sporangia have been dissolved. (**I**) Sporangiospores. (**J**,**K**) Micro-chlamydospores. (**L**) Macro-chlamydospore.

**Figure 4 jof-08-00895-f004:**
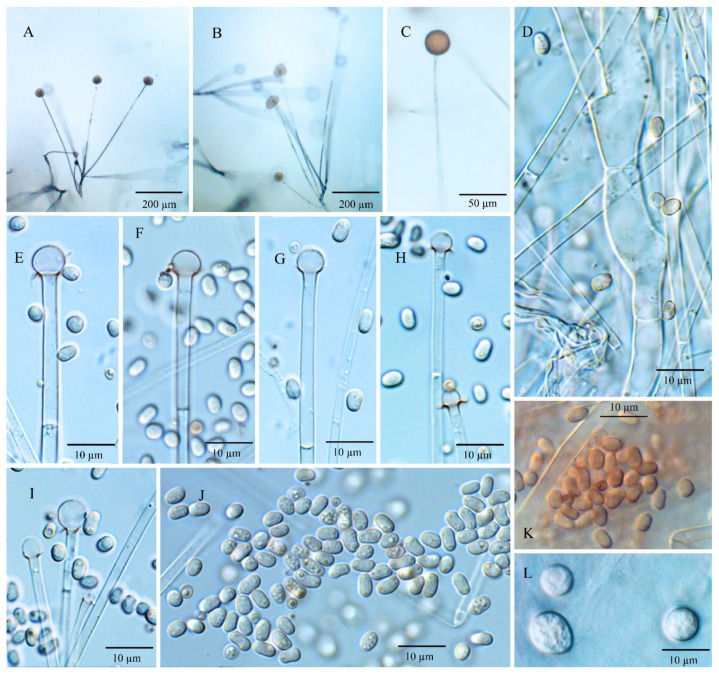
*Umbelopsis macrospora*. (**A**,**B**) Branched sporangiophores. (**C**) Sporangium at tip of sporangiophore. (**D**) Branch point of sporangiophore. (**E**–**I**) Various shapes of collars and columellae at sporangiophore tips after the sporangia have been dissolved. (**J**,**K**) Sporangiospores. (**L**) Micro-chlamydospores.

**Figure 5 jof-08-00895-f005:**
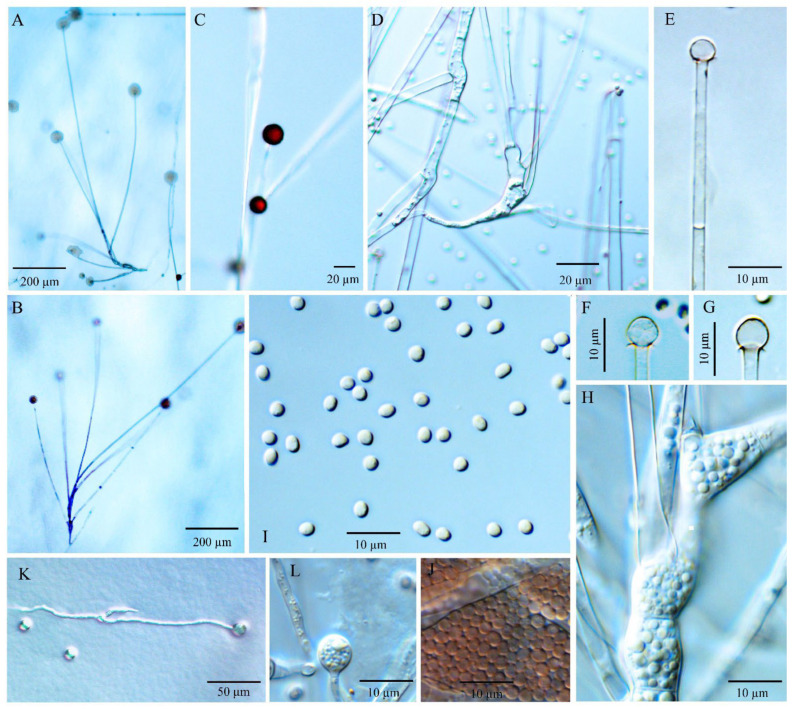
*Umbelopsis microsporangia*. (**A**,**B**) Branched sporangiophores. (**C**) Sporangium at tip of sporangiophore. (**D**,**H**) Branch point of sporangiophore. (**E**–**G**) Various shapes of collars and columellae at sporangiophore tips after the sporangia have been dissolved. (**I**,**J**) Sporangiospores. (**K**,**L**) Micro-chlamydospores.

**Figure 6 jof-08-00895-f006:**
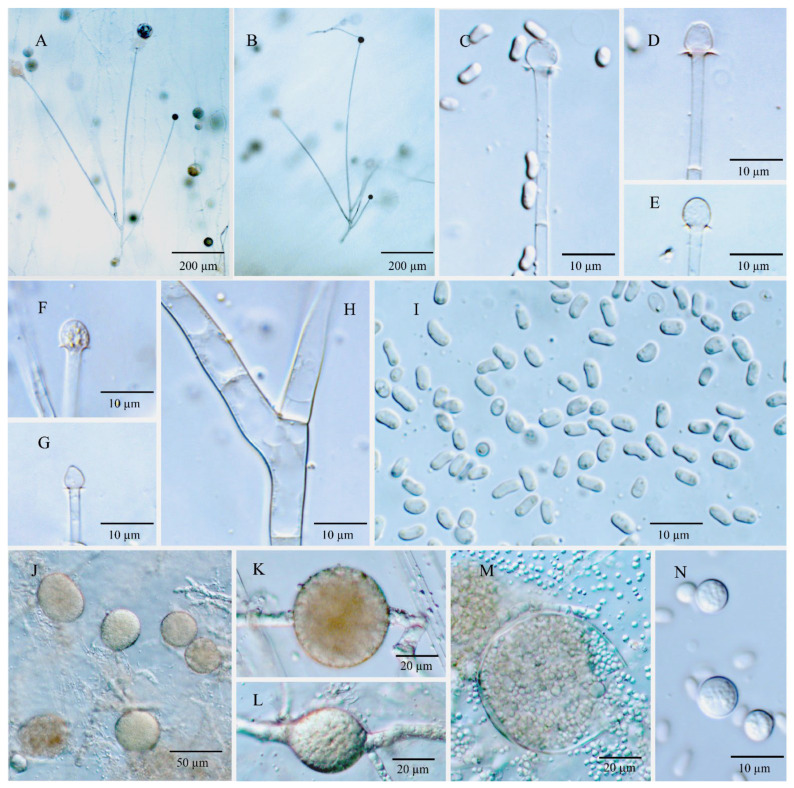
*Umbelopsis oblongielliptica*. (**A**,**B**) Branched sporangiophores. (**C**–**G**) Various shapes of collars and columellae at sporangiophore tips after the sporangia have been dissolved. (**H**) Branch point of sporangiophore. (**I**) Sporangiospores. (**J**–**M**) Various macro-chlamydospores. (**K**). Mature macro-chlamydospores, (**L**). Immature macro-chlamydospores, (**M**). Broken macro-chlamydospore spilling oil droplets. (**N**). Micro-chlamydospores.

**Figure 7 jof-08-00895-f007:**
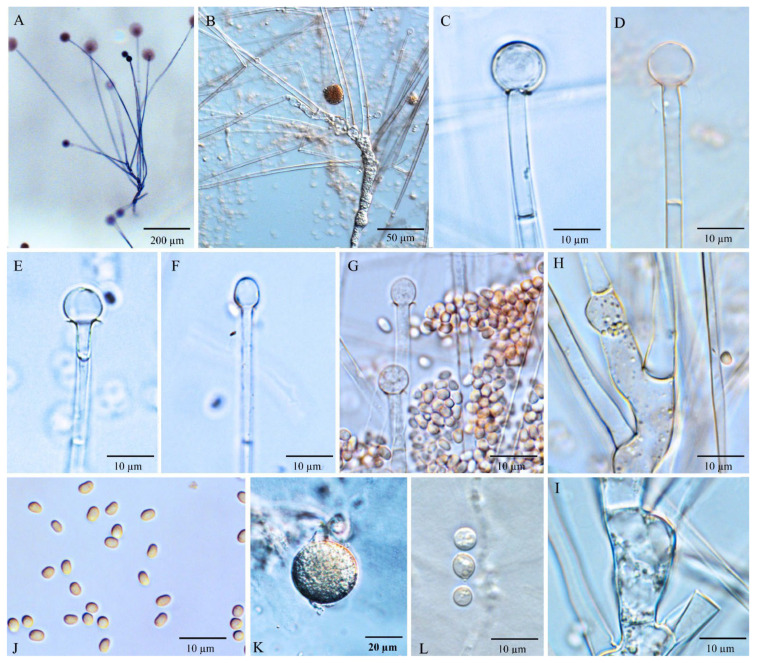
*Umbelopsis ramanniana*. (**A**,**B**) Branched sporangiophores. (**C**–**G**) Various shapes of collars and columellae at sporangiophore tips after the sporangia have been dissolved. (**H**,**I**) Branch point of sporangiophore. (**J**) Sporangiospores. (**K**) Macro-chlamydospores. (**L**) Micro-chlamydospores.

**Table 1 jof-08-00895-t001:** The mainly morphological comparison among *Umbelopsis ramanniana* complex.

Species	Colony Diam. (mm)	Sporangial Diam. (µm)	Sporangial Walls DLQ ^1^	Collars	Columellae (µm)	Sporangiospores (µm)	Macro-Chlamydospores
C1: *U. curvata*	43–48	(11.0–) 15.5–22.5 (–24.5)	rapidly	small but conspicuous	depressed globose (4.0–) 6.0–8.5 (–10) × (3.5–) 5.0–7.0 (–8.5); or subglobose 4.0–6.0 (–7.0)	ellipsoidal and often curving to one side (2.7–) 3.2–4.5 (–5.7) × (1.6–) 2.0–2.4 (–2.8)	abundant on CMA and MEA
C2: *U. ramanniana*	40–44	(11.0–) 15.0–23.0 (–29.5)	rapidly	small but conspicuous	ovoid to oblong-ovoid 4.0–6.0 (–8.0) × (4.5–) 5.0–7.0 (–9.5) µm	ellipsoidal 2.4–3.6 (–4.6) × 1.6–2.4 (–2.6)	less abundant on CMA or MEA
C3: *U. microsporangia*	35–40	(10.0–) 14.0–18.5 (–22.5)	rapidly	small or no	depressed globose (3.0–) 4.0–6.5 (–10.0) × (2.5–) 3.5–5.5 (–9.0); or subglobose (2.5–) 4.0–6.0	ovoid to subglobose (2.4–) 2.6–3.2 (–4.0) × 1.8–2.5 (–2.8)	rare on CMA and less abundant on MEA
C4: *U. dura*	41–45	(10.0–) 15.0–23.5 (–31.5)	slowly	small or no	roundish conical (2.5–) 3.5–6.0 (–7.0) × (3.5–) 4.5–7.0 (–8.0)	ovoid and sometimes slightly narrowing on one end (2.4–) 2.8–3.3 (–4.6) × (1.6–) 1.9–2.4 (–2.8)	undiscovered on CMA and less abundant on MEA
C5: *U. oblongielliptica*	60–62	(11.5–) 15.5–21.5 (–26.0)	rapidly	small but conspicuous	roundish conical 3.5–6.0 × 4–6.5	oblong-ellipsoidal and sometimes slightly curved to one side (3.2–) 4.0–5.0 (–5.5) × 1.6–2.0 (–2.5)	abundant on CMA and MEA
C6: *U. macrospora*	46	(13.5–) 16.0–23.0 (–25.5)	rapidly	small but conspicuous	depressed globose (4.0–) 6.0–8.0 (–10) × (3.5–) 4.0–6.0 (–7.0); or subglobose (3.0–) 5.0–7.0 (–8.0)	oblong-ellipsoidal to ellipsoidal and sometimes slightly curving to one side (3.6–) 4.0–4.9 (–5.5) × (1.7–) 2.4–2.8 (–3.2)	undiscovered on CMA or MEA

Colony diam. and characteristics of the sporangial state were compared on CMA after 10 days at 18 °C. ^1^ DLQ = deliquescent.

## Data Availability

All sequence data are available in NCBI GenBank following the accession numbers in the manuscript.

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
