# Peer review of "The Umbelopsis ramanniana Sensu Lato Consists of Five Cryptic Species"

_jof, 2022, doi:10.3390/jof8090895_

Round 1

Reviewer 1 Report

It is a very good manuscript describing 5 new species from Umbelopsis ramanniana complex. I have sent several corrections and comments as track changes that I believe will improve the quality of the article. I would like to receive the article again after corrections.I found some weaknesses regarding the English language, but as a non-native English speaker, I don't feel comfortable correcting these mistakes. In the supplementary table s1, please indicate the meaning of URM. In the supplementary Figure s1, please correct "subcalde" = subclade.

Author Response

I'm sorry for so many mistakes in the manuscript, and I really appreciate your serious and one-by-one revisions. Your useful feedback has greatly improved our manuscript. The point-by-point response was uploaded as the word file, and the manuscript has now been revised based on the comments. We hope that these comments have been adequately addressed.

Reviewer 2 Report

  • The manuscript entitled "The Umbelopsis ramanniana sensu lato consist of five cryptic  species" implemented strategies for the grouping and revelation of the species from traits that also allowed to distinguish in a better way this species. it was prepared in good quality; also the results and discussion was in-depth.
  • However, a minor revision is suggested:
  • abstract: should indicate the grouping methods used, 
  • Ffigure 1 indicates that the sporangiospores of novel species established  are illustrated on the right side of the tree, but the resolution is not so good and should be improved since they cannot be seen adequately.
  • In the case of table 1, it is recommended to put the units in  the column titles and to use abbreviations in some of the characteristics to minimize the width of the table and facilitate its rapid recognition.
  • The extension of the manuscripts is long because all the figures are requiered to a better recognition but the extension increased, so  it is suggested that Table 2 remain as a supplement since the information is not relevant to the main results obtained, also in the case of the information for each of the species, I believe that the results are very descriptive, consider summarizing or complete the information on a supplementary material,
  • The format style is free and dont require a conclusion section, but is important highligh the contribution of this study.

Author Response

The authors thank the reviewer for the useful feedback to improve our manuscript. The manuscript has now been revised based on the comments, and point-by-point response was uploaded as the word file. We hope that thses comments have been adequatedly addressed.
